# Exploring Dry-Film FTIR Spectroscopy to Characterize Milk Composition and Subclinical Ketosis throughout a Cow’s Lactation

**DOI:** 10.3390/foods10092033

**Published:** 2021-08-29

**Authors:** Amira Rachah, Olav Reksen, Valeria Tafintseva, Felicia Judith Marie Stehr, Elling-Olav Rukke, Egil Prestløkken, Adam Martin, Achim Kohler, Nils Kristian Afseth

**Affiliations:** 1Department of Sustainable Energy Technology, Sintef Industry, Richard Birkelands vei 3, P.O. Box 4760 Torgarden, 7034 Trondheim, Norway; 2Department of Production Animal Clinical Sciences, Faculty of Veterinary Medicine, Norwegian University of Life Sciences, P.O. Box 369 Sentrum, 0120 Oslo, Norway; olav.reksen@nmbu.no (O.R.); adam.martin@nmbu.no (A.M.); 3Faculty of Science and Technology (RealTek), Norwegian University of Life Sciences, P.O. Box 5003, 1432 Ås, Norway; valeria.tafintseva@nmbu.no (V.T.); Achim.kohler@nmbu.no (A.K.); 4Department of Chemistry, Biotechnology and Food Science, Norwegian University of Life Sciences, P.O. Box 5003, 1433 Ås, Norway; stehr89@hotmail.com (F.J.M.S.); elling-olav.rukke@nmbu.no (E.-O.R.); 5Department of Animal and Aquacultural Sciences, Norwegian University of Life Sciences, P.O. Box 5003, 1432 Ås, Norway; egil.prestlokken@nmbu.no; 6Nofima AS, Norwegian Institute of Food, Fisheries and Aquaculture Research, Muninbakken 9-13, Breivika, P.O. Box 6122 Langnes, 9291 Tromsø, Norway; nils.kristian.afseth@nofima.no

**Keywords:** dry-film FTIR spectroscopy, milk, cow health monitoring, subclinical ketosis, fatty acid predictions, PLSR, PCA

## Abstract

The use of technologies for measurements of health parameters of individual cows may ensure early detection of diseases and maximization of individual cow and herd potential. In the present study, dry-film Fourier transform infrared spectroscopy (FTIR) was evaluated for the purpose of detecting and quantifying milk components during cows’ lactation. This was done in order to investigate if these systematic changes can be used to identify cows experiencing subclinical ketosis. The data included 2329 milk samples from 61 Norwegian Red dairy cows collected during the first 100 days in milk (DIM). The resulting FTIR spectra were used for explorative analyses of the milk composition. Principal component analysis (PCA) was used to search for systematic changes in the milk during the lactation. Partial least squares regression (PLSR) was used to predict the fatty acid (FA) composition of all milk samples and the models obtained were used to evaluate systematic changes in the predicted FA composition during the lactation. The results reveal that systematic changes related to both gross milk composition and fatty acid features can be seen throughout lactation. Differences in the predicted FA composition between cows with subclinical ketosis and normal cows, in particular C14:0 and C18:1cis9, showed that dietary energy deficits may be detected by deviations in distinct fatty acid features.

## 1. Introduction

Precision dairy farming involves use of technologies to measure physiological, behavioral, and production traits of individual animals. Several studies have shown that dairy cow diseases often start with subclinical changes in physiological parameters [1,2]. Thus, by using real-time data to monitor individuals to identify deviations, the farmer may intervene before clinical signs become apparent. The use of automatic milking systems (AMS) offers new possibilities for the management of individual cows and the herd. However, sensor technologies that provide actual chemical information of milk and relate this information to the health status of a cow are still lacking [3].

The standard economic parameters of dairy production are milk yield and reproductive performance [4]. However, high milk yields are often associated with reduced reproductive performance and increased incidence of ketosis. Primarily this is because high milk yields increase the risk of a prolonged period of negative energy balance (EB), a period in which energy intake is outstripped by energy requirements from maintenance and lactation. Deeper and more prolonged periods of negative EB negatively impact health and reproductive performance than shorter periods of negative EB. Therefore, it is important to minimize the period a cow is in negative energy balance if dairy production and dairy cow welfare is to be optimized.

Løvendahl et al. [5] found that prediction of EB based on milk fat content, fat/protein ratio and milk yield was not reliable enough for management and decision support in dairy farming. It is known that the fatty acid (FA) composition in milk is affected by variables such as season, herds, nutritional inputs and stage of lactation [6,7]. Stoop et al. [8] documented that milk FA composition is correlated with EB. The negative EB usually seen immediately after calving results in body fat mobilization, which again results in a larger number of long-chained FAs in the milk. The results of Gross et al. [6] showed that particularly changes in 18:1cis-9 and in groups of FAs would be of interest for monitoring EB. Van Haelst [9] documented that cows with subclinical ketosis had a higher proportion of 18:1cis-9 in the milk compared to normal cows, already two weeks before clinical diagnosis of the disease. Martin et al. [10] identified relationships between the proportion of some FAs and onset of luteal activity (OLA) already during the first week after calving. The accuracy of the prediction of early and late OLA was best in the fourth week after calving. A significant relationship between the FAs C14:0, C16:0 and C18:1cis-9 and OLA was observed. The prediction of delayed OLA using FAs gives the farmer a possibility to optimize feeding and treatment of individual cows before delayed OLA occurs [10].

Potential on-farm sensors for milk analysis include near-infrared (NIR) and Fourier-transform infrared (FTIR) spectroscopy. Commercially available sensors for real-time analysis of milk components exist for the former technique [11]. However, FTIR spectroscopy potentially provides better means for exploring subtle chemical distinctions and minor chemical components than NIR. Today, FTIR is widely used in dairy industries for routine analysis for determination of chemical features such as fat, protein, lactose and urea contents [12]. Information on groups of fatty acids and even main single fatty acids can also be available from the spectra [13,14]. Studies also indicate use of FTIR in analysis of features such as protein composition, milk coagulation properties, acidity, and milk minerals [12,15]. Considerable efforts have also been put into the development of FTIR spectroscopy for herd management applications such as EB predictions and cow health monitoring. Several research groups have reported moderate prediction results using FTIR spectroscopy to predict EB, clearly revealing the need for additional efforts to improve prediction qualities [16,17]. Recent studies also show an increasing focus on using FTIR spectroscopy to predict subclinical ketosis, which is related to increased levels of ketone bodies in cow blood, milk, and urine. Negative EB may lead to ketosis, a disease causing reduced appetite, reduced milk yield and weight loss [18,19,20].

Despite research efforts, to the knowledge of the authors, there are currently no commercially available on-farm FTIR based sensors on FTIR in use for cow health monitoring. An overwhelming majority of studies on milk in the field are performed using the liquid transmission approach. This is primarily due to the availability of such instrumentation in the dairy industries. However, other sampling approaches, like dry-film FTIR and attenuated total reflection (ATR) potentially provide better means for exploring detailed chemistry of milk samples [21,22,23,24,25] compared to the liquid transmission approach. In addition, infrared technology is currently being developed at a steady pace, and laser-based infrared technology, e.g., external cavity-quantum cascade laser spectroscopy, is providing new means for obtaining infrared spectra [26]. Thus, the technology of the future on-farm FTIR system has yet to be decided, and there is a need to explore alternative sampling and acquisition technologies. The authors have previously used dry-film FTIR to identify relationships between the proportion of some FAs and OLA already during the first week after calving [10]. Recently, the authors also used dry-film FTIR in combination with milk yield and concentrate intake to improve prediction models for EB [27]. The scope of the present study was to investigate whether dry-film FTIR spectra can be used to relate any systematic changes in milk composition during the first 100 DIM to the appearance of subclinical ketosis in dairy cows. This is to the authors knowledge the first time that dry-film FTIR spectra is used to reveal changes in milk composition throughout lactation.

## 2. Materials and Methods

### 2.1. Production Data

Production data were collected from the research dairy herd at the Animal Production Experimental Centre (SHF) at the Norwegian University of Life Sciences (Ås, Norway) between September 2016 and February 2017. The experiment followed laws and regulations for animal experiments in Norway and was approved by the Norwegian Animal Research Authority. In total 61 Norwegian Red cows were followed from one week before calving until 100 days in milk (DIM). The cows were milked with an AMS (DeLaval International AB, Tumba, Sweden). Milk yield (MY, kg/d) was recorded for each milking and summed to obtain daily yield. For all cows, milk samples were collected thrice weekly. The samples were conserved with bronopol (2-bromo-2-nitropropane-1,3-diol) and stored at 4 °C until subsequent spectroscopic analysis. Samples collected before DIM 5 or after DIM 100 were removed, yielding 2329 samples for data analysis. Details on the production data, feeding and feed analysis have been previously reported [27].

### 2.2. Feeding and Feed Analysis

Production data was obtained from a feeding experiment in which cows were fed grass silage harvested from a timothy (Phleum pratensis) dominated sward ad libitum combined with concentrates. Chemical composition of the feeds is shown in Table 1 [27,28]. Half of the cows were fed silage harvested just before heading of the Timothy grass (early stage of maturity), whereas the other half of the cows were fed a silage from the same crop, harvested 10 days later (normal stage of maturity). In addition, both groups were fed a commercial compound concentrate (FORMEL Energi Premium 70; Felleskjøpet Agri, Lillestrøm, Norge). Concentrates were provided either according to standard feeding templates based on pre-set 304 days lactation yield depending on silage quality and parity according to NorFor [29], or adjusted weekly based on monitored intake of silage and concentrate and requirements for maintenance and production of energy corrected milk (ECM). Silage consumption was automatically recorded through feeding troughs placed on weighing cells (BioControl AS, Rakkestad, Norway) and calculated as described by Kidane et al. [28]. Concentrate intake was recorded as the sum of daily allowances ingested in the AMS and in automatic feeders in the barn.

### 2.3. FTIR Analysis

All 2329 milk samples were analyzed using the dry-film FTIR approach. Prior to dry-film FTIR analysis, all milk samples were removed from refrigeration and placed in room temperature for approximately 30 min. The samples were shaken in a vortex mixer (Whirlimixer, Scientific Industries, Bohemia, NY, USA) for 10 s. The milk samples were diluted with water (75% milk, 25% water) and shaken in a vortex mixer (Whirlimixer) for an 5 additional seconds. Samples (2.5 μL) were then transferred to sample well plates (silicon, 96 wells) and dried at room temperature for approximately one hour. Dry-film FTIR was performed using a high throughput screening eXTension (HTS-XT) unit coupled to a Tensor 27 spectrometer (both from Bruker Optik GmbH, Ettlingen, Germany), equipped with a DLaTGS detector. Spectra were recorded in transmission mode in the spectral region from 4000 to 500 cm^−1^ with a resolution of 6 cm^−1^ and an aperture of 5.0 mm. Background spectra of the silicon substrate were collected before each sample measurement to account for variation in water vapor and CO_2_. All samples were measured in triplicates.

One aliquot of each milk sample was also sent to Tine (Heimdal, Norway) for analysis of bulk chemical composition. The bulk chemical composition analysis was performed using a Bentley FCM 600 FTIR system (Bentley Instruments, Chaska, MN, USA) employing a flow cell for liquid milk analysis. No spectral data were obtained from these analyses, but predictions of fat, protein, lactose, free fatty acids, and urea contents were obtained for all milk samples. For practical reasons, mainly based on the fact that the study was performed during 6 months and that many technicians were involved, some samples were not measured for bulk composition. Thus, a total of 2143 milk samples were subjected to bulk chemical composition analysis.

### 2.4. Data Analysis

A standard quality check was performed on all FTIR spectra, and four outlying samples were removed yielding 2329 spectra to be used for data analysis. Principal component analysis (PCA) and partial least-squares regression (PLSR) were used to explore the dry-film FTIR spectra of milk samples in the study. The FTIR spectra were pre-processed by calculating the second derivative of the spectra using the Savitzky-Golay approach, employing a polynomial of degree 2 and a window size of 13 points in total [30]. Subsequently, normalization of the spectra was performed using extended multiplicative scatter correction (EMSC), employing a linear and quadratic term [31]. The preprocessing was performed before calculating the mean of three replicates [23]. For all data analysis, the wavenumber region between 3200–2800 and 1800–600 cm^−1^ was used. Calibration models for quantification of fatty acids and summed fatty acid parameters were developed using two data sets: (1) Data set A: 219 FTIR spectra from a previously published study were FTIR were used to calibrate for fatty acid contents in milk [23]; and (2) Data set B: 25 milk samples randomly selected from the current study (i.e., selected from the 2329 samples). Milk samples from the calibration data sets were subjected to reference analysis based on gas chromatography, according to previously described protocols [10]. The concentration of individual fatty acids was expressed in percent of total fatty acids present (on a fatty acid methyl ester basis). Summed fatty acid parameters were calculated directly from the GC-FID results: summed saturated fatty acids (SAT), summed monounsaturated fatty acids (MUFA), summed polyunsaturated fatty acids (PUFA). The 244 samples of the calibration data set were combined and used for calibration development in two ways: (1) all 244 samples were used in the calibration data set, and segmented cross-validation, leaving 10% of the samples out in each validation loop. The samples of each validation segment were randomly selected from the whole sample set. (2) The 219 samples from data set A were used to build the calibration, using the 25 samples of data set B as an independent test set. The obtained calibration models using alternative 1 described above were subsequently used to predict fatty acid features of all 2329 milk samples in the current study. All calibration and prediction development were performed using Unscrambler X software (version 10.3, Camo Software AS, Oslo, Norway), and model optimisation (e.g., selection of optimal number of PLS factors) was based on the default criteria provided by the software. 

Classification models differentiating lactation stages DIM 1: 5 ≤ DIM ≤ 50 and DIM 2: 50 < DIM ≤ 100 were established for all samples together and for groups of samples including samples of different parities, i.e., PAR1, PAR2 and PAR > 2 groups, separately. Such an approach in data modelling using hierarchy in the data is known to improve the classification results in linear methods [32]. All parity groups consisted of PAR1 = 895, PAR2 = 608, PAR3 = 343, PAR4 = 326, PAR5 = 81 and PAR6 = 76 samples. The separation into PAR1, PAR2 and PAR > 2 are biologically supported by the fact that cows over the third or fourth parity essentially can be denoted an ‘adult’ cow. Thus, the data split to these three groups had PAR1 = 895, PAR2 = 608, PAR > 2 = 826 samples. Four classification models were established (based on all samples together and on samples of PAR1, PAR2 and PAR > 2 separately) using sparse partial least squares discriminant analysis (SPLSDA) method. SPLSDA method is a variable selection method that allows building robust models [33]. Sparsity parameter and the number of latent variables were optimized using cross validation. Sparsity parameter which defines how many variables that are removed from analysis was optimized in a range between 90% and 99% for each latent variable. Thus, between 1% and 10% of variables were used for each latent variable. The number of latent variables (LVs) were optimized in a range between 1 and 10 LVs. Leave-one individual (cow)-out cross validation was used where at each step of the cross validation all samples of one cow were removed from the data set to establish a model. In a cross validation, samples of each cow were removed once and predicted using samples of all other cows. The cross validation had 22 segments corresponding to 22 cows.

### 2.5. Subclinical Ketosis and β-hydroxybutyrate

The concentration of β-hydroxybutyrate (BHB) in blood was used to diagnose cows with subclinical ketosis. For all cows, blood samples were taken in week 3 of the lactation, and a cow was defined to experience subclinical ketosis if the concentration of β-hydroxybutyrate was more than 1 ng/mL. The measurements of β-hydroxybutyrate were performed at an external laboratory.

## 3. Results and Discussion

### 3.1. Chemical Composition of Milk Samples

The chemical composition of all milk samples was obtained as a reference for the subsequent exploration of dry-film milk analysis. The minimum, maximum, mean and SD of the contents of fat, protein, lactose, urea and FFAs in milk samples are provided in Table 2, revealing expected levels and variations of the chemical constituents of the samples in the study. The corresponding plots of average chemical composition between 5 DIM and 100 DIM (calculated as averages of all cows in the study) showed expected chemical variation (see Appendix A). Fat contents was the chemical component varying most during the lactation period, but a trend of highest fat contents in the beginning and in the end of the lactation is seen, as expected from other studies [34]. Protein content was highest immediately after calving and decreased during the first 30 days of lactation, also as expected from other studies [34,35]. Lactose was the main chemical component with least variation throughout the lactation period, but still a clear maximum of lactose content was reached between 20 DIM and 50 DIM [36]. All in all, the chemical composition analysis shows that milk samples used in the study are consistent with what is expected from related studies in the field.

### 3.2. The Relationship between Dry-Film FTIR Spectra, DIM and Parity

Principal component analysis (PCA) was performed on the dry-film FTIR spectra to investigate if the FTIR spectra could reveal trends in chemical composition during the entire lactation period. PCA was based on the average spectra of all the samples from all cows at one specific DIM, in the period from DIM 5 to DIM 100. The results are presented in Figure 1. The first PC (explaining 68% of the total variation in the FTIR spectra) showed that the FTIR spectra clearly change with respect to DIM. This is illustrated in Figure 1a, where the score value from each DIM is plotted against DIM. Samples from early in the lactation are located to the left in the plot (negative values of PC1) while samples obtained later in the lactation are located to the right (positive values of PC1). The corresponding loading plot (Figure 1b) was used to explore the important spectral bands that were responsible for the patterns observed in the scatter plots. Broadly speaking, the loadings of PC1 show that the main variation in the FTIR spectra is related to changes in bulk composition (i.e., fat, protein and lactose contents). The points that are to the right in the scatter plot (Figure 1a) representing DIM > 50 have higher total lipid content (positive peak at 1751 cm^−1^, i.e., C=O stretching, related to fat), while the points to the left representing DIM < 50 are higher in protein (negative peaks at 1657 and 1549 cm^−1^, i.e., the amide 1 and 2 bands, respectively, related to protein). In addition, the main lactose band at 1074 cm^−1^ (i.e., C-O stretching) follows the same trend as proteins for the PC1. For the subsequent PC’s, no obvious pattern relating the score values to DIM could be found (data not shown). However, to understand more about major spectral variations, it is still relevant to study the respective loadings. For PC2 (explaining 16% of the total variation in the FTIR spectra), shifts in the classical fatty acid region (i.e., 2800–3200 cm^−1^) of the loadings (Figure 1c) suggest that the variation seen is related to changes in fat contents and fatty acid composition. For PC3 (explaining 12.2% of the total variation in the FTIR spectra) a rather complicated mix of bulk chemical components is seen in the loadings (Figure 1d). All in all, the relationship between FTIR spectra and DIM is reflected through a complicated variation of the main and minor chemical components in the milk.

PCA analysis revealed that there is an apparent correlation between DIM and the spectral fingerprints, and that milk samples early in the lactation (low DIM values) can be separated from samples late in the lactation (high DIM values). This makes sense also from a biological point of view: A cow requires a lot of energy in the beginning of the lactation due to the large milk production. Up to approximately 60 DIM, it is not possible for the cow to replace the lost energy just by eating and the cow will experience a negative EB. Thus, grouping the samples into two categories by lactation stages: 5 ≤ DIM ≤ 50 (i.e., DIM 1) and 50 < DIM ≤ 100 (i.e., DIM2) seems to be a good approach for subsequent analysis were also other parameters like parity and fatty acid composition is added. For the current data set, the number of milk samples in each group were as follows: DIM1 = 1273, and DIM2 = 1060 milk samples. An example of the use of lactation stages is provided in Appendix A, where PCA score plots of FTIR milk spectra from three randomly selected cows are plotted. The samples are color coded according to early (i.e., DIM 1) or late (i.e., DIM 2) lactation stage, respectively, and the figure clearly reveals that samples are separated to a good extent, with some overlapping samples. However, interestingly, since the distinction in lactation stage is present along different PCs for the three cows, it is apparent that a huge biological variation on cow level needs to be considered in order to make generic data management systems on individual cow levels based on the current data.

The authors have recently shown that for prediction of EB from FTIR spectra of milk, not only stage of lactation (i.e., DIM, or number of days after calving) is important, but also parity (i.e., the number of times a cow has calved) [27]. In the present data, cows from six different parities were included (i.e., from first- to sixth parity cows). All milk samples from one parity were averaged, and the subsequent PCA analysis are provided in Figure 2. In Figure 2a, the score values of PC 1 (explaining 72.5% of the overall variation) is plotted against the respective parity. Here, a clear separation of the first parity is revealed while parities 2, 3 and 4 are almost stable with gradual increase from group 4 to 6. Interestingly, the corresponding PC1 loading shows that this variation is mainly related to variations in fatty acid composition and fat contents, with main FTIR bands closely resembling the bands appearing in loading plot of PC2 shown in Figure 1c. The driving force of the differences in fat in different parities is reasonable also from a biological point of view. On a general level, one would expect older cows to produce milk with more fat. First parity cows could also generally be expected to experience longer periods of negative EB than older cows, meaning that they also could be expected to produce milk with a lower share of short and medium chain fatty acids.

### 3.3. Prediction of Fatty Acid Features

Fatty acid calibrations based on FTIR milk spectra were obtained for a range of different fatty acids and calculated fatty acid features. The main part of the calibration samples was obtained from a previously published study (i.e., 219 samples, denoted data set A). In addition, 25 milk samples randomly selected from the current study were added as calibration samples (i.e., data set B). In Appendix A, calibration results using data sets A and B together are provided, as well as calibration results using data set A for calibration and data set B as independent test set. The number of LVs were optimized using cross validation. The results clearly reveal that selected fatty acid features are predicted well and within the accuracies of published results [15]. The only calibration models that did not perform well in the conservative test set validation regime chosen, were the CLA and PUFA models developed, as shown in Appendix A. This can most likely be explained by the complexity of the respective calibration models. The minimum, maximum, mean and SD of the predicted fatty acid features in the study is provided in Table 3.

Plotted against DIM, all main fatty acid features revealed systematic changes over time. Overall, average values (all cows) of C10:0, C14:0, C16:0 and the sum of saturated fatty acids increased with DIM (results not shown). In addition, average values (all cows) of C18:0, C18:1cis-9 and sum of monounsaturated fatty acids decreased with DIM (results not shown). To illustrate these trends, C18:0 and C16:0, are presented in Figure 3. The lower plots show how average fatty acid values changes across DIM, corresponding to the trends explained above. These trends are in accordance with the expected biological trends: In early lactation there is a relatively greater negative energy balance triggering mobilization of body fat (adipose tissue) which contains a high proportion of long chain FAs (C18). As the EB becomes more positive less body fat is mobilized and the concentration of long chain FAs in the milk decreases. Concurrently, the improving EB increases de novo synthesis of medium chain fatty acids (C16), an energetically demanding process. The upper plots show corresponding fatty acid trends for three randomly selected cows in the data set. The trends can be recognized for all cows with expected strong individual cow’s variations. 

With fatty acid features predicted for all milk samples, this opens the possibility to explore potential connections between FTIR spectra, fatty acids, DIM and parities. Classification models were built using the SPLSDA method to discriminate DIM1 vs. DIM2 groups of milk samples using FTIR spectra. Based on the scores of this model, the correlation loading plot was made and presented in Figure 4. Such a plot shows the correlation patterns among FAs and groups of FAs, FTIR spectral bands selected by the corresponding SPLSDA model and design parameters such as parity groups and DIM groups (DIM1: DIM ≤ 50, DIM 2: 50 < DIM < 100, PAR1 = first-parity cows, PAR2 = second parity cows, PAR3 = third-parity cows, PAR4 = fourth-parity cows, PAR5 = fifth-parity cows, PAR6 = sixth-parity cows). We can observe from Figure 4 that MUFA, C18:1cis-9, long chain C18:0 and DIM1 are highly correlated to each other. This means that samples of the DIM1 group contain a high proportion of monounsaturated fatty acids in general, and particularly C18:1cis-9. This is in accordance with the trends explained in the previous section and Figure 3. Interestingly, 1645 cm^−1^ and 1545 cm^−1^ bands representing Amide I and Amide II bands, respectively, are strongly correlated with the DIM1 group in the first LV, which means high protein content in DIM1 milk samples. This has the evident biological explanation of high protein content of the first milk in the lactation. Another strong correlation is observed among SAT, C10:0, C14:0, C16:0 and DIM2, all located to the right in the loading plot. This means that samples of DIM2 group contains a high proportion of these three FAs and SAT, also in accordance with the trends explained in the previous section and Figure 3. Interestingly, from these results we can see that in the first 2 LVs there is no indication of strong correlations of parity groups to any of the FAs or groups of DIM. However, the correlations might be seen in the further LVs.

To check whether the observed patterns of correlation of DIM groups to FAs and groups of FAs are the same or differ in each lactation group, similar analysis was done using milk samples corresponding to PAR1, PAR2 and PAR > 2 (i.e., PAR3, PAR4, PAR5, PAR6). Thus, three SPLSDA classification models were established differentiating between DIM1 and DIM2 in each of these groups: PAR1, PAR2 and PAR > 2. The results are presented in Figure 5. Scores of SPLSDA model established using PAR1, PAR2 and PAR > 2 group samples are used to show the correlations in Figure 5a, Figure 5b and Figure 5c, respectively. The correlation patterns are overall very similar to what is observed in Figure 4, but PUFA seems to be much stronger correlated to the DIM1 group in LV2 in PAR1 group samples than in any other groups (see Figure 5a). Interestingly, the 1745 cm^−1^ peak (and other spectral points around 1740–1750 cm^−1^) corresponds to the amount of fat in milk. In all the plots this peak(s) is strongly correlated to DIM2. There is also a biological reason for this: the first milk has more protein, whereas milk later in the lactation period contains more fat necessary for offsprings.

### 3.4. The Relationship between FTIR Spectra and Subclinical Ketosis

In the previous results sections, we have seen how milk composition changes during lactation, and how different milk components are related to each other, DIM and parities. This serves as a chemical validation of the dry-film FTIR approach, but it also provides a platform for interpretation and understanding of dry-film FTIR milk spectra when used for exploring changes in milk composition throughout the lactation period of a cow. The next step is therefore to link the FTIR spectra to important parameters for cattle health monitoring. As stated earlier, EB is a key parameter in dairy cow welfare and performance since deeper and more prolonged periods of negative EB negatively impact health and reproductive performance. EB can be calculated by using the difference between energy intake and energy outputs such as milk, maintenance, pregnancy and growth. But the direct prediction of EB from FTIR spectra is not trivial, as also stated earlier. The present data was recently used for direct prediction of EB [27]. With optimized modelling, including information on milk yield and concentrate intake in addition to the FTIR spectra, results were improved compared to existing literature in the field. However, still less than 50% of the variance of EB in the data set could be explained. This shows that alternative indirect measures of EB of cows for health monitoring purposes will likely be needed for FTIR to be useful in on-farm applications.

Negative EB may lead to ketosis, a disease causing reduced appetite, reduced milk yield and weight loss. In this study, the level of β-hydroxybutyrate (BHB) in blood samples from cows were measured in order to detect cows with subclinical ketosis. Of the 61 cows in the study, 21 cows gave BHB values exceeding 1 ng/mL, which was used as the threshold value for subclinical ketosis. To investigate if FTIR spectra could reveal differences between cows with subclinical ketosis and normal cows, the average of predicted fatty acids was plotted over time for the two groups, respectively. An example of this is shown in Figure 6, where predicted contents of C14:0, C18:0 and C18:1cis9, respectively, is plotted over time for the two respective groups. For C14:0, a clearly lower proportion of C14:0 is found in the beginning of the lactation in milk from cows experiencing subclinical ketosis. For C18:1cis9, this difference is opposite: a clearly higher proportion of C18:1cis9 is found in milk from cows experiencing subclinical ketosis. For both fatty acids, this difference is equalized towards the end of the lactation. This indicates that early detection of subclinical ketosis on cow level is possible by following distinct fatty acid features in milk. C18:0, on the other hand, does not indicate any clear difference between the two groups of cows. Other FAs than the three shown in Figure 6 are likely to follow the same pattern, and whether some FAs are better suited than others, or whether subgroups of FAs is the best choice, is currently an open question. FAs in milk (and other biological systems) are bound to co-vary, meaning that variation in one specific fatty acid will affect the variation of another FA. This aspect has to be taken into account in further research, especially when dealing with spectroscopic calibrations as in the present study [23,38,39].

It is important to note that Figure 6 is based on averages of FAs in milk from normal cows and cows experiencing subclinical ketosis, respectively. Therefore, it is also interesting to investigate if subclinical ketosis can be indicated from FAs of milk from individual cows early in the lactation. Thus, additional PCA were run on all FAs predicted from the FTIR spectra. This was done in three different ways: (1) PCA of FAs from the first milk sample in each lactation (total of 61 samples); (2) PCA of the average of FAs from the three first milk samples in each lactation (total of 61 samples); and (3) PCA of the average of FAs from the five first milk samples in each lactation (total of 61 samples). The averaging of the three or five first milk samples in each lactation was done in order to potentially equal out some of the variance seen in the FA contents from day to day (e.g., as shown in Figure 3). PCA results revealed that analysis of the first milk sample in the lactation did not provide any clear subgrouping of normal cows and cows experiencing subclinical ketosis. However, for the two other PCA alternatives, some clustering of related samples could be seen. In Figure 7, the score plot from PCA of the average of FAs from the five first milk samples in each lactation is provided. In this figure, a further subgrouping of cows related to BHB contents is performed: normal cows (<1.0 ng/mL, 40 cows), subclinical ketosis cows with BHB values between 1.0 ng/mL and 2.0 ng/mL (15 cows), and subclinical ketosis cows with BHB values above 2.0 ng/mL (6 cows). Even though no clear separation is seen in the figure, there is a dominating presence of cows diagnosed with subclinical ketosis on the positive side of PC1, and five of six cows with blood BHB values above 2.0 ng/mL is found in the right side of the plot. The corresponding loadings (data not shown) is explained by the variation between monounsaturated fatty acids (in the positive direction) and saturated fatty acids (in the negative direction). The separation is thereby built on the same variation as provided in the 18:1cis9 plot of Figure 6. It should be expected that a supervised analysis method (e.g., a classification approach) could improve these results, but the present results still show some of the potential in using dry-film FTIR for early detection of subclinical ketosis in single cows. Further research will be needed to find out if milk sampling, instrumental analysis and calibrations can be improved to reduce the day-to-day variations seen in the FTIR spectra, or if also biological variation is an important factor in this variation.

### 3.5. General Discussion

A cheap, rapid, and accurate method for determination of FA composition and other components of milk could be advantageous for monitoring of cows’ health and fertility on farm. This study reveals that dry-film FTIR spectra are suitable for detection and quantification of milk components using PCA analysis and PLSR modelling approaches, both using the FTIR spectra for prediction of fatty acid composition and directly using the spectral fingerprints. The validation of the current approach is first of all obtained through the respective biological interpretations provided. The PCA of average FTIR spectra from all cows in the lactation periods DIM 5−100, DIM 5−50 and DIM 51−100, respectively, shows a clear trend over time with samples from early and late DIM located on opposite sides in the score plots. To maximize the predictive ability of their EB models, Rachah et al. [27] stratified cows based on parity and stage of lactation. Milk production increases rapidly in the start of lactation and peaks around day 55 in milk [40]. Similarly, dry matter intake (DMI) increases in the beginning of lactation typically peaking 50 to 60 DIM [29]. However, DMI increases more slowly that milk yield. Thus, in the first two months of lactation a dairy cow undergoes a period of negative energy balance which affects their metabolism and also the composition of the milk produced [41]. Moreover, excluding milk production the energy requirements and thus EB of cows differs because cattle do not reach their full mature weight before 4 to 8 years of age [42]. The current study also shows that these considerations need to be taken into account when future on farm monitoring systems are developed.

Several studies have reported on the association between FA composition in milk and EB, subclinical ketosis and onset of luteal activity (OLA) [6,9,10]. FAs like C14:0, C16:0, C18:1cis-9 have been reported to have a significant relationship with OLA [10]. C18:1cis-9 and groups of FAs have been suggested as interesting parameters for monitoring of EB [6]. Cows with subclinical ketosis have been reported to produce milk with a higher proportion of C18:1cis-9 than normal cows [9]. Moreover, the origin of C10:0 and C18:0 in cow milk differs, and the proportion of them changes with stage of lactation and thus EB. For improved fertility, it is important to minimize the extent of negative EB through optimal feeding of the cows [43]. Detection of cows with subclinical ketosis enables the farmer to intervene early and avoid reduced cow health and financial loss [9]. Moreover, monitoring EB and early detection of cows likely to experience late OLA makes it easier to give the individual cow optimal treatment and dietary supplements [10]. Thus, the clear possibility to detect subclinical ketosis at an early stage of lactation based on contents of C14:0 and 18:1cis9 (see Figure 6) is an intriguing aspect that needs closer future investigations. Another main factor in this discussion concerns the uncertainty of FA predictions. In the present study, older FA calibrations (published in 2010) were updated with samples from the current experiment, taking place around 10 years later. But still, the test set validation results provided shows that uncertainties are moderately low, with two clear exceptions (i.e., contents of PUFA and CLA). Thus, for future investigations, also the number of calibration samples should be extended to improve prediction uncertainties.

Many of the results of the present study are provided as averages of cows, leading to clear and often smooth trends throughout the lactation period. However, when looking at lactation curves of individual cows (e.g., Figure 3 and Appendix A), the trends are often obscured by larger day-to-day variations. This variation can obviously be related to both prediction and sampling uncertainties, as well as natural biological variation, since every cow has their own unique metabolism. The need for smoothing algorithms to take such variations in lactation curves into account is well known, and similar variations is also apparent from the gross composition analysis included in the current study (see Appendix A). Future on-farm solutions will have to be able to cope with these types of variations, and it is likely that advanced data processing approaches like machine learning will be a natural part of future development in this direction.

The results of the present study clearly shows some of the potential that lies in using FTIR, and particularly dry-film FTIR, for cow health monitoring. It is known from previous studies [22,23,24] that dry-film analysis increases the sensitivity towards minor components. Since water is removed before analysis, the main confounding component is out of play, making it possible to discover interrelation between the remaining relevant components more clearly. However, this also means that the direct link between the FTIR spectra and the gross composition (i.e., fat, protein and lactose contents in percentage of sample contents) is lost [22]. Thus, it is still open for discussion which sampling method should be used in future on-farm applications, since there is no real knowledge on what sensitivity is needed for specific applications. A direct comparison between dry film and liquid analysis is currently lacking and will be important for future development. Dry-film analysis will require a sampling step (drying) that might take around 30 min (air drying). The whole procedure of sampling and dry-film analysis can easily be automated using robotics [44]. In addition, the speed is not always of the essence in this application, and sampling minor components like fatty acids should be well covered if it happens on once-daily basis. Representative milk sampling, however, is of major importance, and needs further investigations.

## 4. Conclusions

This study illustrates the feasibility of using dry-film FTIR spectroscopy for characterization and quantification of milk components during the lactation period of cows. The results reveal that systematic changes related to both gross milk composition and fatty acid features can be seen throughout lactation. It is also shown that FTIR predictions of particular fatty acids early in the lactation period can potentially be used to predict cows diagnosed with subclinical ketosis later in the lactation period. A direct comparison between sampling approaches and infrared technologies will be important in order to develop future on-farm solutions for cattle health monitoring based on infrared spectroscopy.

## Figures and Tables

**Figure 1 foods-10-02033-f001:**
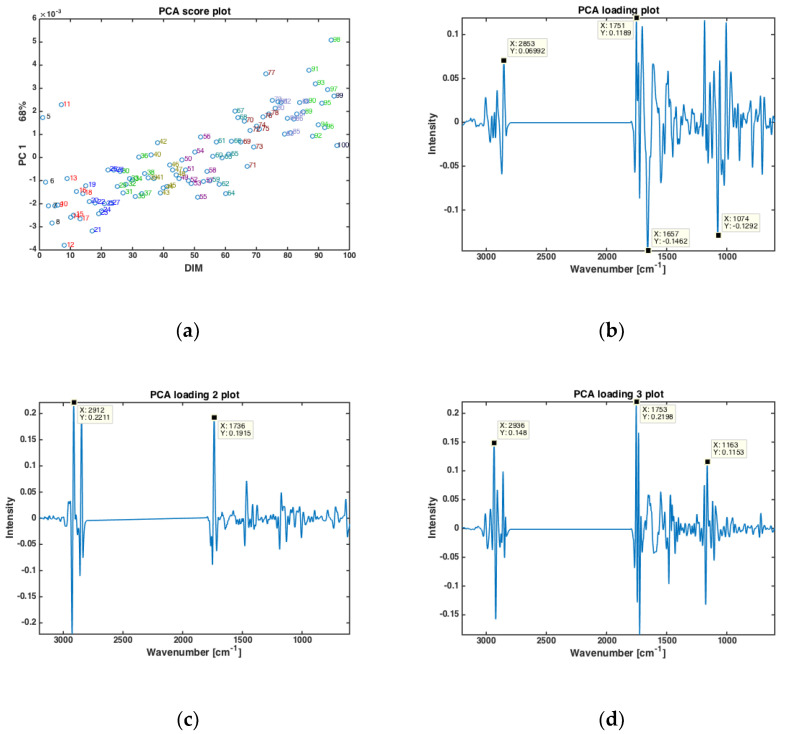
PCA of all FTIR spectra of milk samples averaged at a given DIM (**a**) Score plotted against DIM, and loading plots of (**b**) PC 1, (**c**) PC2 and (**d**) PC3.

**Figure 2 foods-10-02033-f002:**
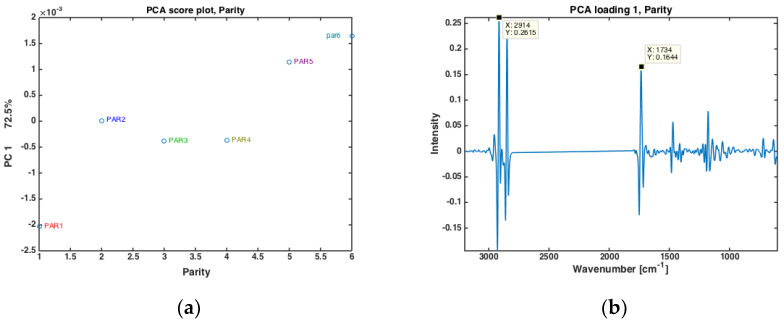
PCA of all FTIR spectra of milk samples for different parity groups (PAR1 = first-parity cows, PAR2 = second parity cows, PAR3 = third-parity cows, PAR4 = fourth-parity cows, PAR5 = fifth-parity cows, PAR6 = sixth-parity cows) (**a**) Score plot, and (**b**) loading plot of PC 1.

**Figure 3 foods-10-02033-f003:**
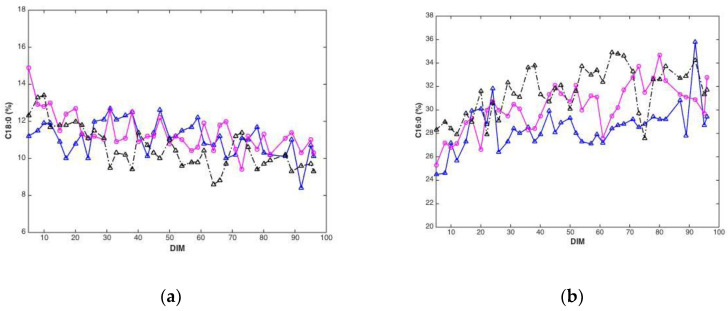
Fatty acid content (percentage of total weight of fatty acids) C18:0 (**a**) and C16:0 (**b**) during 5–100 DIM for three individual cows, and calculated as the average C18:0 (**c**) and C16:0 (**d**) of all cows at a given DIM.

**Figure 4 foods-10-02033-f004:**
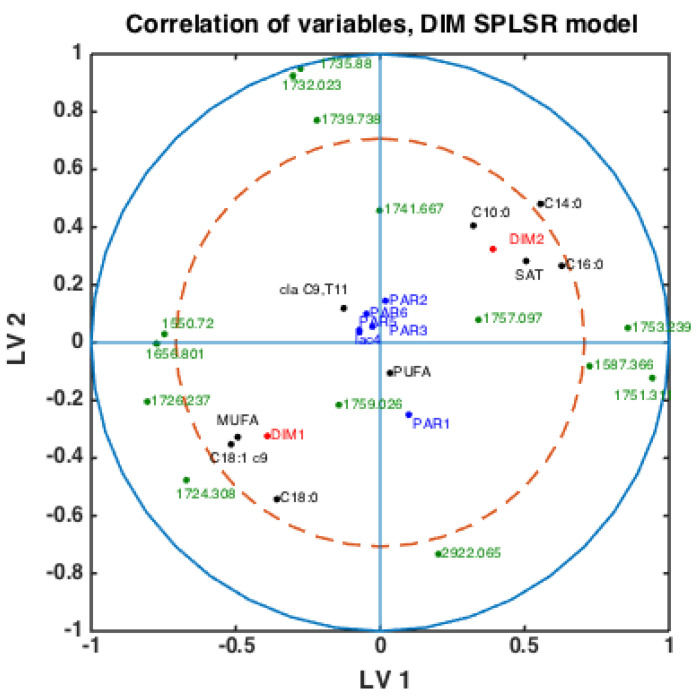
Correlation loading plot obtained from SPLSDA model for classification of FTIR spectra into the two main groups of DIM: DIM1: DIM ≤ 50; DIM2: 50 < DIM < 100 including all samples of all parities.

**Figure 5 foods-10-02033-f005:**
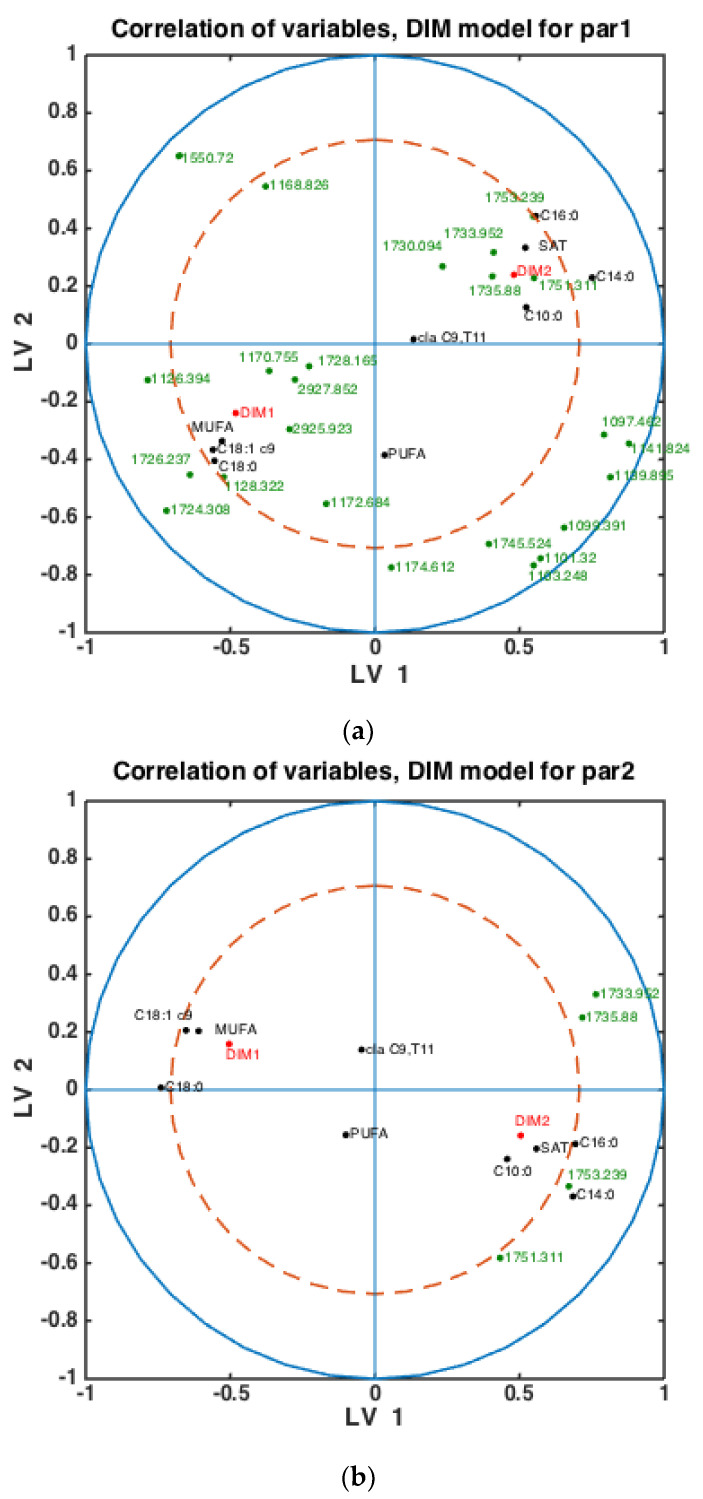
Correlation loading plots obtained from SPLSDA models for classification of FTIR spectra into the two main groups of DIM: DIM1: DIM ≤ 50; DIM2: 50 < DIM < 100. Models obtained using samples of groups PAR1 in (**a**), PAR2 in (**b**) and PAR > 2 in (**c**) are used to generate the correlation plots.

**Figure 6 foods-10-02033-f006:**
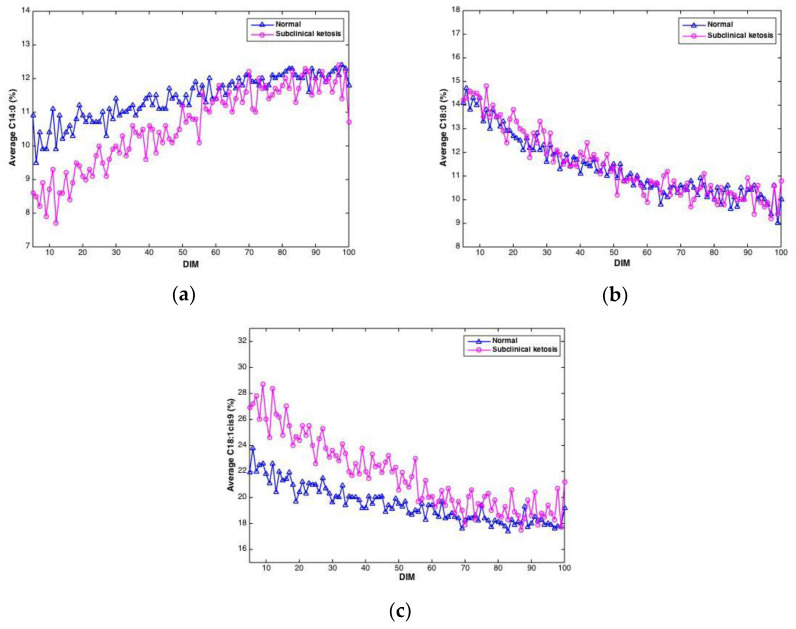
Average contents of C14:0 (**a**), C18:0 (**b**) and C18:1cis9 (**c**) in milk samples from cows experiencing subclinical ketosis (red) and normal cows (blue).

**Figure 7 foods-10-02033-f007:**
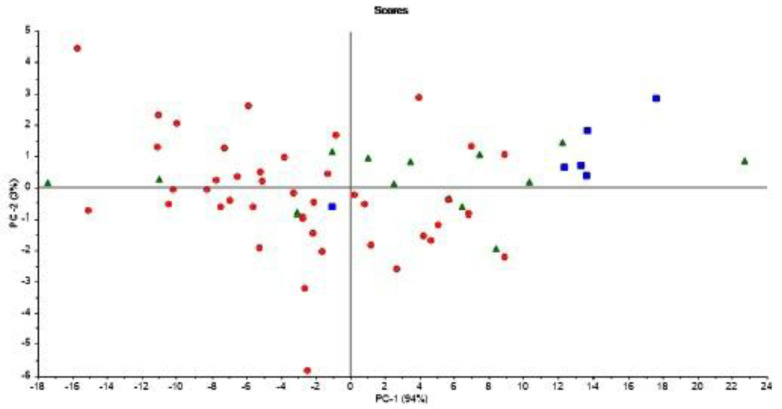
Score plot from PCA of the average of FA’s from the five first milk samples in each 61 lactation in the study. Samples are coded according to normal cows (<1.0 ng/mL, circles), subclinical ketosis cows with BHB values between 1.0 ng/mL and 2.0 ng/mL (triangles), and subclinical ketosis cows with BHB values above 2.0 ng/mL (squares).

**Table 1 foods-10-02033-t001:** Mean values of chemical composition and nutritional values of feeds used (±SD). All units in g/kg dry matter if not otherwise specified.

	Early Harvest	Normal Harvest	Concentrate
Dry matter (g/kg feed)	261 ± 23.5	359 ± 10.2	878
Ash	75 ± 2.5	62 ± 1.9	78
Crude protein	151 ± 6.9	127 ± 5.8	277
Crude fat	31 ± 1.9	27 ± 1.9	65
aNDFom ^1^	579 ± 7.1	614 ± 7.6	179
iNDF ^2^ (g/kg NDF)	134 ± 59	206 ± 32	206
Starch			289
OMD ^3^ (%)	80.3 ± 1.6	71.9 ± 5.6	
NEL20 ^4^ (MJ/kg DM)	6.75 ± 0.0	6.18 ± 0.0	7.35
AAT20	80.5 ± 2.1	77.0 ± 2.8	163
PBV20	42.5 ± 2.1	9.0 ± 24.0	41

^1^ aNDFom = amylase treated and residual ash corrected Neutral Detergent Fiber; ^2^ iNDF = indigestible aNDFom; ^3^ OMD = organic matter digestibility; ^4^ NEL20, AAT20, PBV20 = Standard feed values for net energy lactation (NEl), amino acids absorbed (AAT) and rumen protein balance (PBV) calculated at 20 kg dry matter intake (DMI) as described by the NorFor system (Volden, 2011).

**Table 2 foods-10-02033-t002:** Minimum, maximum, mean and standard deviation (SD) of parameters measured for 2143 milk samples from 61 cows [37].

Chemical Component	Min	Max	Mean	SD
Fat (%) ^1^	2.0	8.0	4.0	1.0
Protein (%) ^1^	2.5	4.7	3.4	0.3
Lactose (%) ^1^	4.1	5.5	4.9	0.2
Urea (mmol/L)	2.4	8.3	5.1	0.8
FFAs ^2^ (mmol/L)	0.1	5.4	0.5	0.4

^1^ Percentage of milk weight; ^2^ FFAs = Free fatty acids.

**Table 3 foods-10-02033-t003:** Minimum, maximum, mean and standard deviation (SD) of the predicted fatty acid content in 2329 milk samples from 61 cows. All values are expressed as percentage of total weight of fatty acids.

Chemical Component	Min	Max	Mean	SD
C10:0	0.5	6.1	3.1	0.7
C14:0	4.1	23.0	11.1	1.6
C16:0	16.0	43.5	27.9	3.1
C18:0	3.7	22.1	11.5	1.9
C18:1cis-9	7.1	39.4	20.3	3.6
CLA ^1^	0	1.1	0.5	0.1
SAT ^1^	46.4	76.8	67.5	4.0
MUFA ^1^	14.9	46.2	24.9	3.9
PUFA ^1^	1.4	4.9	2.3	0.3

^1^ CLA = Conjugated linoleic acid; SAT = sum of saturated fatty acids; MUFA = sum of monounsaturated fatty acids; PUFA = sum of polyunsaturated fatty acids.

## Data Availability

Not applicable.

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
