# Peer review of "Exploring Dry-Film FTIR Spectroscopy to Characterize Milk Composition and Subclinical Ketosis throughout a Cow’s Lactation"

_foods, 2021, doi:10.3390/foods10092033_

Round 1

Reviewer 1 Report

General comment:

The present manuscript is a very comprehensive study on the use of FTIR-spectroscopy for monitoring health of cows and herd. The authors have recently published the use of this technique to estimate the energy balance (EB) from milk as well. The scientific work has been carried out in a consistent and ordered manner and the manuscript is very well written in general.

I would only like to point some minor issues I have found in the manuscript that might help the authors:

Introduction:

  • Lines 58-59: I would suggest changing “milk fat content, fat:protein ratio” to “milk fat content, fat/protein ratio”
  • Line 82: change “milk coagulation properties and acidity, and milk” to “milk coagulation properties, acidity, and milk”

Materials and methods:

  • Lines 153-154: what is the speed at which the sample is vortexed?
  • Line 160 and all along the manuscript: please put cm-1 with the “-1” as super index.
  • Line 168: what do you mean here when saying that “Due to some missing samples and failed measurements”? Please explain this.
  • Line 177: state what EMSC means here.
  • Lines 181-224: align (justify) text properly.

Tables:

Table 1: please state here the meaning of NDF

Reviewer 2 Report

  • If this work is part of a Master’s thesis, the original document deserves at least to be cited here. It seems that Ms. Stehr contributed more to this manuscript than what is mentioned in “contribution” section.
  • In introduction give references, and in discussion discuss the advantages of this method over other more conventional and well stablished methods for fatty acid profiling (GC-FID).
  • I do not see the applicability of this method for monitoring “cattle health”. Clarify in the text with more data and details on its potential implementation. Or adapt the title, for instance mentioning subclinical ketosis but not health in general.
  • Table 1 presents mean values and SD? Clarify. Why concentrate does not include SD?
  • Improve the quality of figures.

Round 2

Reviewer 2 Report

If this work is part of a Master’s thesis, the original document deserves to be cited here.